# Proposing Specific Neuronal Epithelial-to-Mesenchymal Transition Genes as an Ancillary Tool for Differential Diagnosis among Pulmonary Neuroendocrine Neoplasms

**DOI:** 10.3390/genes13122309

**Published:** 2022-12-07

**Authors:** Tabatha Gutierrez Prieto, Camila Machado Baldavira, Juliana Machado-Rugolo, Eloisa Helena Ribeiro Olivieri, Eduardo Caetano Abilio da Silva, Alexandre Muxfeldt Ab’ Saber, Teresa Yae Takagaki, Vera Luiza Capelozzi

**Affiliations:** 1Laboratory of Genomics and Histomorphometry, Department of Pathology, University of São Paulo Medical School (USP), São Paulo 01246-903, SP, Brazil; 2Health Technology Assessment Center (NATS), Clinical Hospital (HCFMB), Medical School of São Paulo State University (UNESP), Botucatu 18618-970, SP, Brazil; 3International Center of Research/CIPE, AC Camargo Cancer Center, São Paulo 01509-900, SP, Brazil; 4Molecular Oncology Research Center, Barretos Cancer Hospital, Barretos 14784-400, SP, Brazil; 5Fundação Oncocentro do Estado de São Paulo (FOSP), São Paulo 05409-012, SP, Brazil; 6Division of Pneumology, Instituto do Coração (Incor), Medical School of University of São Paulo, São Paulo 01246-903, SP, Brazil

**Keywords:** pulmonary neuroendocrine neoplasms, epithelial-to-mesenchymal transition, morphology, diagnosis, metastasis

## Abstract

Pulmonary neuroendocrine neoplasms (PNENs) are currently classified into four major histotypes, including typical carcinoid (TC), atypical carcinoid (AC), large cell neuroendocrine carcinoma (LCNEC), and small cell lung carcinoma (SCLC). This classification was designed to be applied to surgical specimens mostly anchored in morphological parameters, resulting in considerable overlapping among PNENs, which may result in important challenges for clinicians’ decisions in the case of small biopsies. Since PNENs originate from the neuroectodermic cells, epithelial-to-mesenchymal transition (EMT) gene expression shows promise as biomarkers involved in the genotypic transformation of neuroectodermic cells, including mutation burden with the involvement of chromatin remodeling genes, apoptosis, and mitosis rate, leading to modification in final cellular phenotype. In this situation, additional markers also applicable to biopsy specimens, which correlate PNENs subtypes with systemic treatment response, are much needed, and current potential candidates are neurogenic EMT genes. This study investigated EMT genes expression and its association with PNENs histotypes in tumor tissues from 24 patients with PNENs. PCR Array System for 84 EMT-related genes selected 15 differentially expressed genes among the PNENs, allowing to discriminate TC from AC, LCNEC from AC, and SCLC from AC. Functional enrichment analysis of the EMT genes differentially expressed among PNENs subtypes showed that they are involved in cellular proliferation, extracellular matrix degradation, regulation of cell apoptosis, oncogenesis, and tumor cell invasion. Interestingly, four EMT genes (*MAP1B*, *SNAI2*, *MMP2*, *WNT5A*) are also involved in neurological diseases, in brain metastasis, and interact with platinum-based chemotherapy and tyrosine–kinase inhibitors. Collectively, these findings emerge as an important ancillary tool to improve the strategies of histologic diagnosis in PNENs and unveil the four EMT genes that can play an important role in driving chemical response in PNENs.

## 1. Introduction

The current classification of pulmonary neuroendocrine neoplasms (PNENs) maintains the four major histotypes of the previous WHO 2015 [1], including tumors with some specific histologic and immunophenotype patterns, but presents bimodal behavior: gradual proliferation in typical carcinoids (TCs) that are low-grade neuroendocrine malignancies (NET G1) and infrequently metastasize; intermediate proliferation in atypical carcinoids (ACs) that are intermediate-grade malignancies (NET G2); the large cell neuroendocrine carcinomas (LCNECs); and the small cell lung carcinomas (SCLCs) that are high-grade malignancies (NEC G3) which rapidly metastasize with worse prognosis [2]. Currently, the histologic diagnostic approach applied to PNENs is based on morphologic criteria first reported in WHO 1999 [3], including a mitotic index, the presence of necrosis, and cell size, whereas a Ki67 proliferation index is not included in the assessment for classification purposes. Although this nomenclature is not harmonized with the classification proposed by WHO/IARC [4], a significant overlap exists. On the one hand, PNEN classification and diagnosis, particularly for TCs and ACs, are complicated by many issues, such as an inconstant natural history and generic symptoms [5]. Surgical resection remains the only curative option for TCs and ACs, but there is no agreement between PNEN management procedures concerning the best treatment approaches in the recurrence/metastatic scenario [6]. On the other hand, the pathologist has experienced routine cases in which the precise discrimination between TC and AC, or AC and LCNEC is not achieved employing the classical morphological parameters, thereby suggesting that ancillary tests are needed to complete effective clinical management. For example, the rate of mitosis to classify AC ranges from 2 to 10 mitoses per 2 mm^2^. However, if the number of mitoses is greater than 10 per 2 mm^2^, the diagnosis will be LCNEC; however, there are cases with a high rate of mitosis that maintain the carcinoid morphology. Should there be a grade 3 pulmonary neuroendocrine tumor (PNET) (>10 per 2 mm^2^) or should we classify these as LCNEC but note a carcinoid-like morphology and the mitotic rate/proliferation index? Although it is known that these criteria have several restrictions, the diagnostic approach remains unaltered in the recent WHO 2021 classification [2], which may result in important challenges for clinicians’ decisions.

Pulmonary NENs have a prevalence of between 15% and 20% in all lung cancers. Because of the low frequency of carcinoid TC (1–2%), AC (0.2%) and LCNEC (1–3%) [7], these tumors are termed “orphan diseases” [8]. They represent a heterogeneous group of neoplasms from the endocrine cells derived from the neuroectoderm, neural crest, and endoderm [9]. Epithelial-to-mesenchymal transition (EMT) is a dynamic process that is activated in several types of tumors. In neuroendocrine tumor cells, EMT-related genes can be involved in the behavior of these tumors, enhancing the cell’s motility and invasiveness, reducing cell death by apoptosis, resistance to chemotherapeutic drugs, etc. [9,10,11]. In this context, additional markers also applicable to biopsy specimens, which correlate PNENs subtypes with systemic treatment response, are much needed, and current potential candidates are neurogenic EMT genes. Previously, using cDNA microarrays, we evaluated the expression profile of 17 EMT-related genes among PNENs histological types, expressing different levels of (*BMP1*, *BMP7*, *CALD1*, *CDH1*, *COL3A1*, *COL5A2*, *EGFR*, *ERBB3, PLEK2*, *SNAI2*, *STEAP1*, *TCF4*, *CDH2*, *KRT14*, *CAV2*, *DSC2*, and *IL1RN*) and which were also related to mechanical barriers [12]. These results prompted us to examine whether a classification of PNENs according to EMT molecular subtyping, in conjunction with classical morphologic characteristics, may provide a clinically relevant solution. Therefore, we performed a reanalysis of that gene expression profile, aiming to identify differentially expressed mRNA with a potential diagnostic value between PNETs and PNECs. In the current study, we observed that fifteen EMT genes were differentially expressed among the PNENs, including (*MMP3, ITGAV*, *KRT14*, *PLEK2*, *SNAI2*, *GSK3B*, *ITGB1*, *MAP1B*, *TCF3*, *VPSA13*, *SMAD2*, *MMP2*, *SPARC, WNT5A*, and *ITGA5*), among which three genes were also selected in our previous study allowing for the discrimination of TC from AC (*SNAI2*), LCNEC from AC (*KRT14*, *PLEK2*, and *SNAI2*), and SCLC from AC (*PLEK2* and *SNAI2*). Interestingly, four EMT genes (*MAP1B*, *SNAI2*, *MMP2*, and *WNT5A*) were also involved in neurological diseases, brain metastasis, and response to platinum-based chemotherapy or tyrosine–kinase inhibitors. Collectively, these findings emerge as an important ancillary tool to improve the strategies of histologic diagnosis in PNENs and unveil the four neuronal EMT genes that can play an important role in driving brain metastasis in PNENs.

## 2. Material and Methods

### 2.1. Patients and Samples (Discovery Cohort)

We identified 24 patients at A. C. Camargo Cancer Center, in São Paulo, Brazil, and Hospital do Amor, in Barretos, Brazil, who were surgically resected with PNENs [10 carcinoid tumors (5 TC and 5 AC)], 4 LCNEC, 10 SCLC, and had fresh-frozen tissue available from their primary tumor.

Two pathologists (T.G.P. and V.L.C) carried out a blinded comprehensive review of all tumors to confirm their histological subtype according to the WHO 2021, the mitotic count, the presence of an organoid pattern (rosettes, pseudo rosettes, palisading, spindle cells) and necrosis [2]. Patient demographics and clinicopathological characteristics were obtained from medical records and included age, sex, smoking history, tumor size, tumor stage (according to the International Association for the Study of Lung Cancer classification system, 8th edition), and follow-up information [1]. The internal ethics committees of all the participating institutions approved this study’s protocol (process number 1.077.100) with a waiver for informed consent by their review boards.

### 2.2. Gene Expression Profile Data

The neoplastic area was micro-dissected during the frozen section procedure to ensure the inclusion of neoplastic tissue and distant non-neoplastic tissue as control. Total mRNA was extracted from fresh-frozen tumors and normal tissues using the QIAsymphony miRNA CT 400 kit (Qiagen, CA, USA) according to the manufacturer’s instructions. RNA integrity and quality were determined using the Bioanalyzer 2100 (Agilent Technologies). Complementary DNA was synthesized using the cDNA—RT² First Strand Kit (Qiagen Sample & Assay Technologies) according to the manufacturer’s protocol. The difference of expression in EMT genes was evaluated by the real-time PCR method. Quantitative reverse transcription–polymerase chain reaction (qRT–PCR) was performed using the RT² Profiler PCR Array System (PAHS-090Z; Qiagen, Dusseldorf, Germany) kit for the human epithelia-to-mesenchymal transition (EMT) pathway with 84 target genes. The array includes a total of 84 EMT genes, 5 housekeeping genes (*ACTB*, *B2M*, *GAPDH*, *HPRT1*, *RPLP0*), 1 genomic DNA control (GDC) to assess contamination, 3 reverse transcriptase controls (RTC) that certify the efficiency of the reverse transcription step, and 3 positive PCR controls (PPC) consisting of an artificial DNA sequence certifying the test accuracy. Each 96-well plate includes SYBR^®^ Green-optimized primer assays for a thoroughly researched panel of 84 EMT genes. Furthermore, the high-quality primer design and RT2 SYBR^®^ Green qPCR Mastermix formulation enable the PCR array to amplify 96 gene-specific products simultaneously under uniform cycling conditions. The samples were amplified using Applied Biosystems Step One Plus (Applied Biosystems, Carlsbad, CA, USA). The cycling conditions were as follows: 95 °C for 10 min, 40 cycles at 95 °C for 15 s, 60 °C for 1 min, followed by a dissociation period. The data were then analyzed in the StepOne software (v. 2.0, Applied Biosystems, Life Technologies Corporation (Carlsbad, Califórnia, EUA) using the Δ threshold cycle (Ct) method (2−ΔΔCt) [13]. All data were normalized by the housekeeping genes, and normal lung tissue specimens were used as case control. The Ct cutoff was set to 35, and Fold–Change (FC) cutoff was set to ≥2.0. The calculation of normalization for both normal and tumor tissue was the same, as follows: normalized target gene signal data of tumor samples (test) and lung normal tissue (control); [target gene signal normalized to housekeeping genes; e.g., 2^^−ΔCt^, where −ΔCt = −(Ct_Target − Avg. Ct_HKG)]. The studied genes that were differently expressed (DEG) were obtained from a panel of 84 EMT genes, the same platform that was applied in a previous study by our group (GSE181381) [12].

### 2.3. Functional Enrichment Analysis of EMT Genes

To further elucidate the function and signaling pathways involved in the enrichment of EMT-related genes, we inputted the genes that were differentially expressed (DEGs) among PNENs histological subtypes, totaling 15 genes, into Metascape (https://metascape.org/, accessed on 15 July 2022) [14] to perform Gene Ontology (GO) function, KEGG, and REACTOME pathway analyses. The GO analysis was composed of 3 categories, namely, biological processes (BPs), cellular components (CCs), and molecular functions (MFs). Results that met the threshold value with *p* < 0.05 were regarded as significant.

### 2.4. PPI Network Construction and Module Analysis (Validation Cohort)

To reveal the functional interactions among the proteins encoded by these genes, the selected genes were uploaded onto STRING (https://string-db.org/, accessed on 15 July 2022) tools to map their PPI network [15]. Results that presented a significant combined interaction score of *p* > 0.9 were considered significant.

### 2.5. Identification of Chemicals Associated with EMT-Related Genes

The CTD database (https://ctdbase.org/, accessed on 10 August 2022) is a major public resource of literature-based, artificially planned linkages between chemistry, genetic products, phenotypes, diseases, and environmental exposure [16]. We searched the database for the EMT genes to identify chemicals associated with neoplasia and brain disorders.

### 2.6. Data Management and Statistical Analysis

Data were collected and managed using REDCap electronic data capture tools hosted at A. C. Camargo Cancer Center, in São Paulo, Brazil. Considering the non-normal distribution of our data, all statistical tests employed in this study to examine the difference between categories and groups were non-parametric tests. The chi-square test or Fisher’s exact test, the non-parametric Kendall tau-b correlation coefficient, and Spearman’s rank correlation coefficient were used to examine differences in categorical variables, whereas the Kruskal–Wallis test and Mann–Whitney U test were used to detect differences in continuous variables between groups of patients. We used the Statistical Package of Social Science (SPSS) version 18 (SPSS, IBM Corporation, Armonk, New York, USA) for all statistical analyses and RStudio 2022.02.0 software (RStudio, PBC, Boston, MA, USA, https://www.rstudio.com, accessed on 20 September 2022) to construct a dot plot graphic. All tests with *p* < 0.05 were deemed statistically significant and a Bonferroni correction was used when necessary.

## 3. Results

### 3.1. Samples and Clinicopathological Data

Twenty-four fresh-frozen tumor specimens were obtained from patients diagnosed with stage I–IV PNENs. The tumor samples were enriched for earlier stages (I/II) in carcinoid tumors according to the 8th edition of the International Union for Cancer Control (UICC) TNM Classification of Malignant Tumors [1] and consisted of primary lung tumors obtained by surgical resection. The TC and AC cases were selected to obtain a balanced number of cases in each group, as in our previous study [12]. The cohort comprises 5 TC, 5 AC, 4 LCNEC, and 10 SCLC according to the current World Health Organization (WHO) 2021 classification [2]. As expected, a male patient with a tobacco history was more associated with SCLC when compared to LCNEC and carcinoid tumors, with statistical significance. The pathological stage showed a difference between carcinoid tumors and carcinomas. While all patients with carcinoid tumors (AC and TC) were in an early stage, those with carcinomas (SCLC and LCNEC) were in an advanced stage (*p* < 0.01). The median age of the patients was 56 years (range: 33–76 years). Table 1 summarizes the clinical characteristics of patients, stratified by histological types.

### 3.2. The Expression of EMT-Related Genes in PNENs Subtypes: Implication on Morphology

The Mann–Whitney U test was used to evaluate the difference in EMT gene expression among PNENs histological subtypes. Fifteen EMT genes (*MMP3*, *ITGAV*, *KRT14*, *PLEK2*, *SNAI2*, *GSK3B*, *ITGB1*, *MAP1B*, *TCF3*, *VPSA13*, *SMAD2*, *MMP2*, *SPARC*, *WNT5A*, and *ITGA5*) were differently expressed, mainly in SCLC. Furthermore, among these EMT genes, four genes (*MMP2*, *MAP1B*, *SNAI2*, and *WNT5A*) were involved in neurologic diseases (Table 2). A dot plot graphic was constructed to show the distribution of these gene expressions among histological subtypes (Figure 1).

To analyze the impact of these EMT genes on the morphological features of PNENs, and concerning the difficulty of diagnosis between these entities, we separated the analysis according to these groups: SCLC/LCNEC, AC/TC, LCNEC/AC, and SCLC/AC (Figure 1).

#### 3.2.1. SCLC vs. LCNEC

Six EMT genes (*GSK3B*, *ITGB1*, *MAP1B*, *TCF3*, *VPSA13*, and *SMAD2*) were differently expressed between SCLC and LCNEC and may be involved in the difference in morphological patterns of these PNENs subtypes. The expression of these genes was higher in SCLC than in LCNEC, and with the exception of *ITGB1* and *TCF3*, all genes were also overexpressed when compared to normal lung tissue.

#### 3.2.2. AC vs. TC

Two genes (*MMP3* and *SNAI2*) were differently expressed regarding the carcinoid tumor. In TC patients, *MMP3* was highly expressed, while *SNAI2* was under-expressed despite being more expressed than in AC.

#### 3.2.3. LCNEC vs. AC

Five genes (*MMP3*, *ITGAV*, *KRT14*, *PLEK2*, and *SNAI2*) were differently expressed between LCNEC and AC. Patients with LCNEC presented higher expression of these genes when compared to AC patients.

#### 3.2.4. SCLC vs. AC

In this group, seven genes (*PLEK2*, *MAP1B*, *MMP2*, *SNAI2*, *SPARC*, *WNT5A*, and *ITGA5*) were differently expressed between SCLC and AC. Patients with SCLC presented high expression of these genes when compared to the AC patients.

### 3.3. Functionally Biological Enrichment Analysis

Functionally, the 15 EMT-related genes (*MMP3*, *ITGAV*, *KRT14*, *PLEK2*, *SNAI2*, *GSK3B*, *ITGB1*, *MAP1B*, *TCF3*, *VPSA13*, *SMAD2*, *MMP2*, *SPARC*, *WNT5A,* and *ITGA5*) were involved in several biological pathways associated with cellular proliferation, extracellular matrix degradation, regulation of cell apoptosis, mutation burden with the involvement of chromatin remodeling genes, mitosis rate, and tumor cell invasion, leading to modification in final cellular phenotype. In the Metascape analysis, the GO produced a list of top-level biological processes and a heatmap of enriched terms related to the input genes, which included: “Epithelial to mesenchymal transition in colorectal cancer”, “Tissue morphogenesis”, “Burn wound healing”, “Establishment of cell polarity”, “Factors and pathways affecting insulin-like growth factor (IGF1)-Akt signaling”, “PID MYC repress pathway”, “Negative regulation of cell differentiation” (Figure 2A). We then consulted the enrichment analysis in DisGeNET, where these 15 EMT-related genes seem to be involved in several diseases, as shown in Figure 2B. Finally, Figure 3A,B shows the network of enriched terms: (A) colored by cluster-ID, where nodes that share the same cluster-ID are typically close to each other; (B) colored by *p*-value, where terms containing more genes tend to have a more significant *p*-value.

### 3.4. PPI Network Construction and Module Analysis (Validation Cohort)

Firstly, the PPI network with the proteins that are encoded by these genes using the STRING database was performed, which included the 15 functional partners with the highest interaction score, namely (*MMP3*, *ITGAV*, *KRT14*, *PLEK2*, *SNAI2*, *GSK3B*, *ITGB1*, *MAP1B*, *TCF3*, *VPSA13*, *SMAD2*, *MMP2*, *SPARC*, *WNT5A,* and *ITGA5*). Then, to analyze the interaction between these EMT-related genes on the morphological features of PNENs, we selected four groups: SCLC vs. LCNEC (*GSK3B*, *ITGB1*, *MAP1B*, *TCF3*, *VPSA13*, and *SMAD2*), AC vs. TC (*MMP3* and *SNAI2*), LCNEC vs. AC (*MMP3*, *ITGAV*, *KRT14*, *PLEK2*, and *SNAI2*), and SCLC vs. AC (*PLEK2*, *MAP1B*, *MMP2*, *SNAI2*, *SPARC*, *WNT5A*, and *ITGA5*). For all analyses, the edges represent protein–protein associations that contribute to a shared function and involve only those proteins with a high edge score (confidence ≥ 0.9). Thus, we amplified these proteins’ molecular organization, including other important molecules that interact in this network which can be visualized as a network of differentially connected nodes shown, as shown in Figure 4A–E.

### 3.5. Chemicals Associated with EMT Genes: Link to Neoplasms and Brain Metastasis

Through the Comparative Toxicogenomics Database—CTD database—among the 15 EMT genes described, four genes (*MMP2*, *MAP1B*, *SNAI2*, and *WNT5A*) were related to neoplasms and brain disorders, named neuronal EMT-related genes. Different chemicals have been identified that affect the expression of these genes and we highlighted those presented with an antineoplastic effect. The complete results are shown in Table 3.

## 4. Discussion

Molecular and protein profiles to classify PNENs have been evaluated in recent years [17,18]. Notably, in PNENs, no driver mutations have been identified. The most frequent molecular abnormality is a mutation in the *MEN-1* gene (5%), which is associated with a poorer prognosis [19]. Gene expression analysis of relapsed PNENs identified chromosomal rearrangements, and the markers MET, TES, and STK39 were found to be highly upregulated in relapsed tumors, although these genes have not been validated [20]. OTP gene, a transcription factor, has been suggested as a putative molecular marker to distinguish aggressive from less aggressive PNENs, on the basis of gene and protein expression profiling [21]. However, the antibody against the OTP protein is only obtainable as a polyclonal antibody for immune staining; thus, further studies are required to discover a stable monoclonal antibody. A recent molecular classification for SCLC includes NEUROD1, ASCL1, POU2F3, and YAP1 (NAPY) [22], which may allow personalized treatment [23]. However, at the protein level, evaluation of the NAPY classification appears more complicated and further investigations are necessary to determine if the NAPY classification is also appropriate for LCNEC with clinical implications.

Genomic and transcriptomic analyses of SCLC and LCNEC also indicate an important overlap with common (biallelic) inactivation of *TP53* and *RB1*, as recently reported by Lantuejoul and colleagues [23]. In LCNEC, an SCLC subtype with *RB1* and *TP53* inactivation is recognized as having a low neuroendocrine gene expression profile, with low *ASCL1* and high *NOTCH* gene expression (referred to as type II LCNEC). Almost 40% of LCNECs have molecular alterations often identified in non-small cell lung cancer (*STK11*, and *KEAP1* mutations) with high expression of *ASCL1* and neuroendocrine markers (defined as type I LCNEC) [23,24]. Remarkably, these type I LCNECs generally have a functioning wild-type *RB1* gene [23]. A different chemotherapy response has been correlated with *RB1* gene status in high-grade NENs [25] but not in all studies [26]. Thereby, a classification of PNENs according to molecular and protein profiles in conjunction with classical morphologic characteristics still represents a challenge for clinical decisions.

cDNA microarray analysis has shown that phenotypic differences at the cellular level are associated with differences in the presence, absence, and abundance of particular RNAs. Our previous finding that differential expression of 17 EMT-related genes was involved in epithelium desmosomes assembly, and cell motility for invasion and metastases [12], led us to speculate that these factors may modify the tumor cell phenotype to act as a barrier receptor. However, in that study, we did not determine whether the EMT process involved in the PNENs cells may be useful to differentiate specifically between TC vs. AC, AC vs. LCNEC, and AC vs. SCLC. In the present study, we re-evaluated the micro-array dataset GSE181381 to verify whether phenotypic differences at the cellular level are associated with differences in the EMT gene expression.

Under the conditions of this study, EMT genes presented a differential spectrum of expression that may be associated with PNENs phenotypes. They included: (1) overexpression of *GSK3B*, *ITGB1*, *MAP1B*, *TCF3*, *VPSA13,* and *SMAD2* in SCLC compared to LCNEC; (2) overexpression of *MMP3* in TC and under-expression of *SNAI2* in AC; (3) overexpression of *MMP3*, *ITGAV*, *KRT14*, *PLEK2,* and *SNAI2* in LCNEC compared to AC; (4) *PLEK2*, *MAP1B*, *MMP2*, *SNAI2*, *SPARC*, *WNT5A,* and *ITGA5* overexpression in SCLC compared to AC. Functionally, these genes are associated with the involvement of chromatin remodeling genes, mitosis rate, and tumor cell invasion, leading to modification in the final cellular phenotype. Moreover, a strong interaction was obtained between these EMT-related genes and the matched proteins. Among these 15 EMT genes, *MMP2*, *MAP1B*, *SNAI2*, and *WNT5A* stand out for their relationship with neoplasms and brain disorders, and are thus termed neuronal EMT genes, and with different chemicals, including platinum-based chemotherapy and tyrosine–kinase inhibitors, which affect the expression of these genes and have been identified with an antineoplastic effect.

Given the continuous clinical requirement to perform a diagnosis on small biopsy tissue specimens, the increasing information about the molecular biology of PNENs, and the increasing systemic treatment options, a classification that is treatment outcome-related and applicable in a biopsy specimen is essential for the practicing pathologist. Considering that PNENs originate from the neuroectodermic cells [27,28,29], it is reasonable to assume that epithelial-to-mesenchymal transition (EMT) genes involved in the genotypic transformation of neuroectodermic cells lead to modification in final cellular phenotype. EMT has been divided into three types based on the biological events during which it occurs: type 1 EMT is associated with developmental processes, such as gastrulation and neural crest cell migration, type 2 EMT with wound healing, and type 3 EMT with tumor progression and metastasis [30,31]. In the present study, there was overexpression of *MMP3* and *SNAI2* in TC compared to AC. *MMP3* is known to manipulate cell phenotype and promote tumor invasion in hepatocellular carcinoma [32]. *MMP3* has also been detected in the nucleus of the cell to control nuclear events, such as apoptosis and cancer progression. In the nucleus, *MMP3* function is associated with the induction of apoptosis, based on the premise that apoptotic cells overexpress *MMP3* in comparison with normal cells [33]. Another property of *MMP3* is related to the induction of the cleavage of nuclear proteins such as PARP, as well as other proteins that contribute to DNA repair and mRNA processing, leading to an increase in DNA damage, which will eventually induce apoptosis [33]. *SNAI2* (Slug) is an EMT transcription factor of the SNAI family and is known to be involved in type 2 EMT during epithelium repair [34], suggesting that the partial EMT of epithelial cells is mediated by *SNAI2*, which in turn is mediated by the epidermal growth factor receptor [35,36]. Indeed, our results, in concordance with Galvan and colleagues’ study, support the notion that the expression of *Snai2* transcription factor protein in PNENs is involved in EMT, with an inverse correlation with E-cadherin expression [37]. In agreement with the Galvan study, by transmission electron microscopy, we showed that desmosomes disassembly barrier intensity was the expression phenotype factor that coincided with the level of CDH2 adherent junctions’ proteins, associated with the risk of lymph node and distant metastasis in PNENs [12].

While we found the higher expression of *MMP3* in TC vs. AC, despite the lower aggressiveness of the former tumor type, the higher expression of *SNAI2* may justify some TC cases with favorable pathologic features following an unpredictable unfavorable clinical course. Therefore, *MMP3* and *SNAI2* may represent ancillary tools to improve the histologic classification of TC, which is particularly problematic due to of frequent overlap with AC on one arm of the diagnostic spectrum and LCNEC or SCLC on the other arm of the spectrum [38]. Functionally, both genes are associated with extracellular matrix degradation, mutation burden with involvement of chromatin remodeling genes, mitosis rate, and tumor cell invasion, leading to modification in the final cellular phenotype. In this condition, we can infer that both EMT genes bring more information than recognizing mitoses on hematoxylin and eosin staining, which may be hindered by the inhomogeneous distribution of the chromosomes and in criteria used for distinguishing them from nuclei karyorrhexis due to chromatin disintegration [39]. Moreover, interobserver subjectivity for TC versus AC is strongly induced by the small number of mitoses required for a panty classification, thereby making every single count essential [39]. Another concern is determining the cellular proliferation index by Ki-67, which is not considered a diagnostic protagonist in PNENs grading, probably due to an overlap in cutoff values separating TC from AC, and because collinearity between Ki-67 proliferation index and mitotic grading was observed [40], thus limiting additional prognostic information beyond the standard morphologic criteria [3]. In addition, the mitotic number, and the presence of comedonecrosis can be underestimated in biopsies. In matched biopsy and resection specimen analysis of PNENs, a Ki-67 cutoff greater than 20% in the biopsy was appropriate for separating low-grade PNENs from high-grade PNENs with 100% sensitivity and specificity [41]. Hitherto, Ki-67 proliferation index cannot be used to separate typical from atypical carcinoids. Furthermore, in LCNEC, Ki-67 proliferation index may be less than 20% [42]. Thus, the current classification advises diagnosing carcinoid tumors on biopsy specimens as “carcinoid tumor not otherwise specified”. Furthermore, a preoperative biopsy diagnosis is usually not sufficient to support a firm decision on the extent of surgery. In the current study, *MMP3* and *SNAI2* EMT genes are also involved in cellular proliferation, and, according to the levels of expression, may represent a complementary tool to differentiate TC from AC.

In the present study, we also found high levels of *KRT14* EMT gene expression in LCNEC compared to AC, as well as high levels of *MAP1B* in SCLC. These EMT genes are involved in phenotypic changes related to the dimension and stability of the cytoskeleton during tumorigenesis [43]. Their expression may minimize the known difficulty to separate LCNEC from SCLC, as well as AC from LCNEC and AC from SCLC in terms of cell dimension [44], with significant interobserver variation and biological similarities contributing to this aspect [45]. Quantitative measurements of PNENs have disclosed an important overlap of cell size in SCLC versus LCNEC, suggesting at least a subjective criterion to separate these entities [46]. Moreover, the cellular characteristics of SCLC may be inhomogeneous in greater specimens; consequently, a percentage of cells of SCLC may present a larger cell size, and some SCLCs combined with adenocarcinoma or squamous cells carcinoma further complicate the assessment [47]. Reflecting the clinical–pathologic problems stated hitherto, the use of morphologic criteria for a merely histologic differentiation of LCNEC from SCLC appears to prove inadequate for providing an objective and clinically pertinent classification, particularly when applied in small biopsy specimens.

The STRING database analysis confirmed the PPI network of these EMT-related genes, its molecular organization as a network of differentially connected nodes, and dynamic synergism. Firstly, we include in the analysis all fifteen EMT-related genes that were differentially expressed among PNENs. Then, we perform the PPI network analysis in four distinct groups: SCLC vs. LCNEC (*GSK3B*, *ITGB1*, *MAP1B*, *TCF3*, *VPSA13*, and *SMAD2*), AC vs. TC (*MMP3* and *SNAI2*), LCNEC vs. AC (*MMP3*, *ITGAV*, *KRT14*, *PLEK2*, and *SNAI2*), and SCLC vs. AC (*PLEK2*, *MAP1B*, *MMP2*, *SNAI2*, *SPARC*, *WNT5A*, and *ITGA5*). Importantly, the function of these EMT genes was associated with cell proliferation, cell motility, matrix degradation, tumor invasion, antiapoptotic activity, and tumor spreading [48,49].

Last but not least, assessing the Comparative Toxicogenomics Database (CTD), we found that *MMP2, MAP1B, SNAI2,* and *WNT5A* EMT genes interact with platinum-based chemotherapy and tyrosine–kinase inhibitors. Higher *MMP2* expression was detected in epithelial ovarian cancer sensitive to chemotherapy [50]. Cisplatin properties involve a reduction in the migration and invasiveness of cancer cells, which has been shown to be related to decreased *MMP2* activity [51]. *MAPB1* gene belongs to a large family of proteins involved in microtubule assembly, which is an essential step in stabilizing microtubules [52]. Disrupting microtubule dynamics is one of the most successful and widely considered targets of cancer chemotherapy agents [53,54]. The action of chemotherapeutic agents that stabilize or destabilize microtubules is regulated by intracellular proteins, including tumor suppressors, oncogenes, and microtubule motor proteins. Disruption of microtubules by treatment with microtubule-targeting drugs triggers p53-dependent apoptosis [53].

SNAI family members were found to directly repress estrogen receptors [55] and enhance the anti-apoptotic behavior of cancer cells, contributing to resistance to therapy [56]. Other studies support the role of EMT in maintaining cancer stem cells, which can be inherently resistant to treatment [57]. The evolving role of *SNAI2* as a modulator of resistance in breast cancer has motivated attention to therapeutic strategies based on reversing EMT to prevent tumor progression and re-sensitizing tumor cells to endocrine therapy [58].

*WNT5A* gene is a tumor suppressor gene for various cancers but a protooncogene for prostate cancer [59]. Chemotherapy or radiation promotes Wnt signaling and protects proliferating cells from cell cycle arrest or apoptosis [60].

Our study has limitations regarding the value of these genes as biomarkers and their real clinical applicability. To be used as biomarkers in individual cases, gene expression levels should have a validated cut-off point defined in large independent series and need to have a standardized methodological approach. Furthermore, to consider only the median of the gene expression value is not enough to discriminate different subtypes of PNENs; however, these markers could be useful as an ancillary tool for the histological diagnostic. Therefore, to validate our discovery cohort, we assessed an in silico approach and showed the PPI network of these correspondent proteins.

Overall, optimization of histologic diagnosis in PNENs might be achieved by a combination of *MMP2, MAP1B, SNAI2,* and *WNT5A* EMT gene expression as a complementary tool to the current WHO classification dependent on further validation in a large cohort and assessment of the protein expression of related molecules. They may represent additional markers applicable to biopsy specimens which may also correlate carcinoid and carcinomas subtypes with systemic treatment response. Although they are not therapeutic targets at this time, routine inclusion of the four EMT genes’ expression may expand the effects of these genes and determine whether there is an interaction with treatment. However, new studies may be required in order to validate the true differences and differential predictive effects of neuronal EMT genes in PNENs. In this emergent scenario, for example, PARP inhibition through DNA damage mechanism can be implemented as a diagnostic and predictive biomarker for the selection of patients for personalized treatments.

## Figures and Tables

**Figure 1 genes-13-02309-f001:**
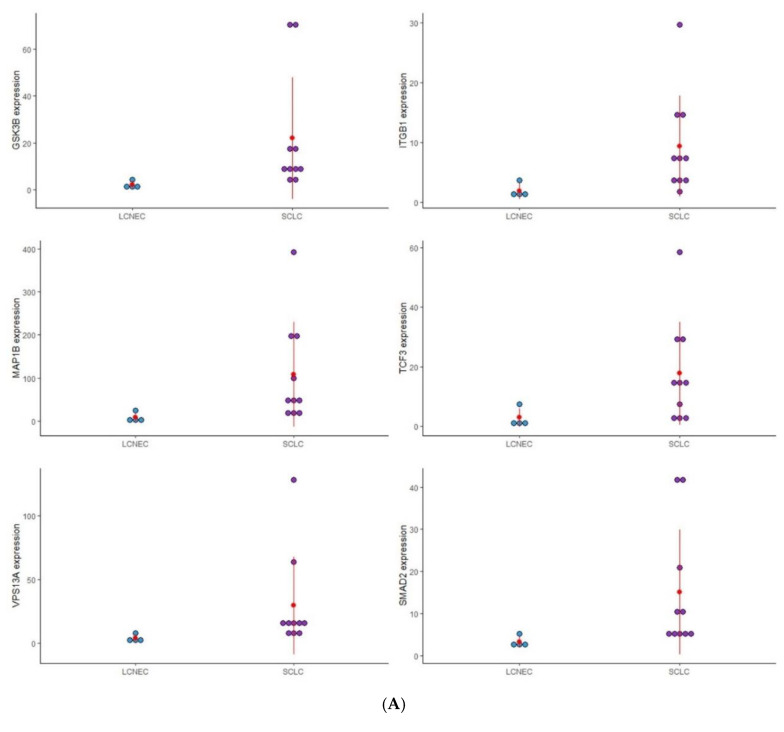
Dot plot graphic to display the distribution of EMT gene expression among the four distinct groups of PNENs histological subtypes. (**A**) Expression of (*GSK3B*, *ITGB1*, *MAP1B*, *TCF3*, *VPS13A*, and *SMAD2*), between SCLC and LCNEC; (**B**) Expression of (*MMP3*, *ITGAV*, *KRT14*, *PLEK2*, and *SNAI2*), between LCNEC and AC; (**C**) Expression of (*MMP3* and *SNAI2*), between AC and TC; (**D**) Expression of (*PLEK2*, *MAP1B*, *MMP2*, *SNAI2*, *SPARC*, *WNT5A*, and *ITGA5*), between SCLC and AC.

**Figure 2 genes-13-02309-f002:**
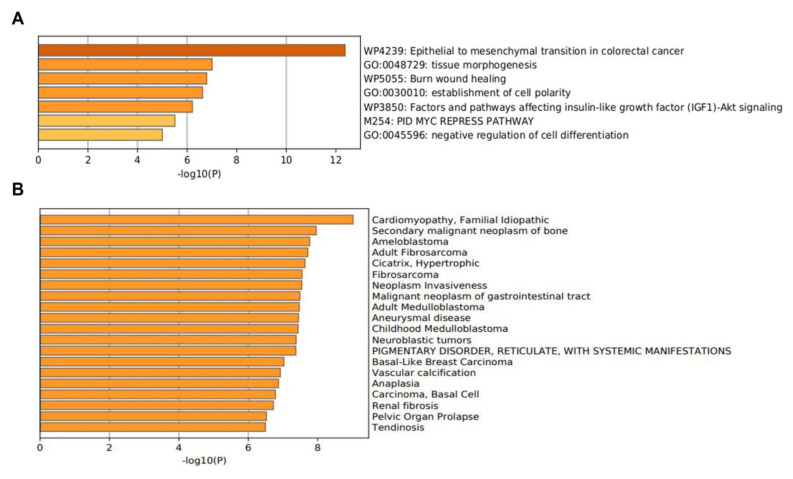
The enrichment analysis of 15 EMT-related genes (*MMP3, ITGAV, KRT14, PLEK2, SNAI2, GSK3B, ITGB1, MAP1B, TCF3, VPSA13, SMAD2, MMP2, SPARC, WNT5A,* and *ITGA5*) using Metascape. (**A**) Heatmap of enriched terms colored by *p*-values. (**B**) The enrichment analysis in DisGeNET, where 15 EMT-related genes seem to be involved in several diseases.

**Figure 3 genes-13-02309-f003:**
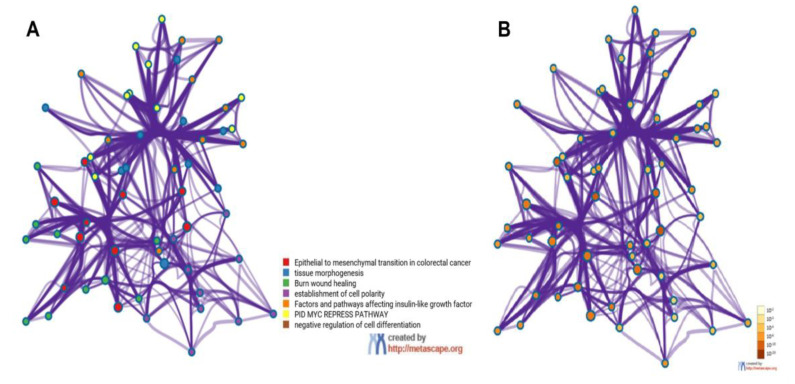
Metascape cluster analysis across input gene lists (*MMP3, ITGAV, KRT14, PLEK2, SNAI2, GSK3B, ITGB1, MAP1B, TCF3, VPSA13, SMAD2, MMP2, SPARC, WNT5A,* and *ITGA5*), Network of enriched terms: (**A**) colored by cluster-ID, where nodes that share the same cluster-ID are typically close to each other; (**B**) colored by *p*-value, where terms containing more genes tend to have a more significant *p*-value.

**Figure 4 genes-13-02309-f004:**
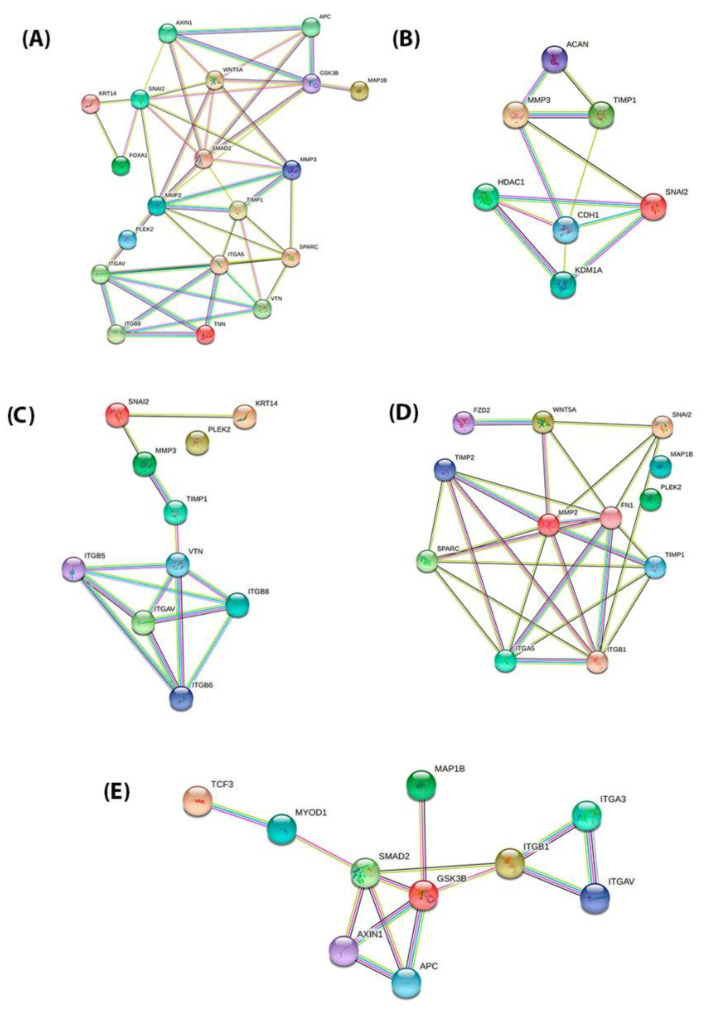
PPI network of the STRING tools among the proteins encoded by 15 EMT genes and divided into four distinct groups. The combined interaction score of *p* > 0.9 was considered significant. (**A**) PPI network of the 15 EMT genes (*MMP3*, *ITGAV*, *KRT14*, *PLEK2*, *SNAI2*, *GSK3B*, *ITGB1*, *MAP1B*, *TCF3*, *VPS13A*, *SMAD2*, *MMP2*, *SPARC*, *WNT5A*, and *ITGA5*) (**B**) PPI network of the AC vs. TC (*MMP3* and *SNAI2*); (**C**) PPI network of the LCNEC vs. AC (*MMP3*, *ITGAV*, *KRT14*, *PLEK2*, and *SNAI2*); (**D**) PPI network of the SCLC vs. AC (*PLEK2*, *MAP1B*, *MMP2*, *SNAI2*, *SPARC*, *WNT5A*, and *ITGA5*); (**E**) PPI network of the SCLC vs. LCNEC (*GSK3B*, *ITGB1*, *MAP1B*, *TCF3*, *VPS13A*, and *SMAD2*).

**Table 1 genes-13-02309-t001:** Demographic and clinicopathologic characteristics of 24 patients diagnosed with PNENs.

Characteristics	Histological Types			
	SCLC (N = 10)	LCNEC (N = 4)	AC (N = 5)	TC (N = 5)
**Gender**				
Male	6 (60.0%)	1 (25.0%)	1 (20.0%)	3 (60.0%)
Female	4 (40.0%)	3 (75.0%)	4 (80.0%)	2 (40.0%)
**Age (median, yrs)**				
<56	3 (30.0%)	1 (25.0%)	4 (80.0%)	3 (60.0%)
≥56	7 (70.0%)	3 (75.0%)	1 (20.0%)	2 (40.0%)
**Smoke Status**				
Yes	10 (100%)	3 (75.0%)	1 (20.0%)	1 (20.0%)
No	0 (0.0%)	1 (25.0%)	4 (80.0%)	4 (80.0%)
**Prophylatic Radiotherapy**				
Yes	3 (30.0%)	1 (25.0%)	0 (0.0%)	0 (0.0%)
No	7 (70.0%)	3 (75.0%)	4 (100%)	3 (100%)
**Stage at tumor analysis**				
**^a^** **T Stage †**				
T1	0 (0.0%)	0 (0.0%)	2 (66.7%)	1 (33.3%)
T2	1 (16.7%)	2 (50.0%)	0 (0.0%)	1 (33.3%)
T3	0 (0.0%)	0 (0.0%)	1 (33.3%)	0 (0.0%)
T4	5 (83.3%)	2 (50.0%)	0 (0.0%)	1 (33.3%)
**^a^** **N Stage †**				
N0	0 (0.0%)	1 (25.0%)	3 (100%)	2 (66.7%)
N1	0 (0.0%)	1 (25.0%)	0 (0.0%)	1 (33.3%)
N2	2 (33.3%)	1 (25.0%)	0 (0.0%)	0 (0.0%)
N3	4 (66.7%)	1 (25.0%)	0 (0.0%)	0 (0.0%)
**^a^** **M Stage †**				
M0	4 (40.0%)	2 (66.7%)	3 (100%)	3 (100%)
M1	6 (60.0%)	1 (33.3%)	0 (0.0%)	0 (0.0%)
**Pathological stage**				
*Early stage (I/II)*	0 (0.0%)	0 (0.0%)	5 (100%)	5 (100%)
*Advanced stage (III/IV)*	10 (100%)	4 (100%)	0 (0.0%)	0 (0.0%)
**Systemic Chemotherapy**				
Yes	8 (80.0%)	3 (75.0%)	2 (50.0%)	2 (66.7%)
No	2 (20.0%)	1 (25.0%)	2 (50.0%)	1 (33.3%)
**Status**				
Alive	0 (0.0%)	1 (25.0%)	4 (80.0%)	5 (100%)
Dead	10 (100%)	3 (75.0%)	1 (20.0%)	0 (0.0%)
**Follow-up (months)**	30 (1–168)			

PNENs: pulmonary neuroendocrine neoplasms; TC: typical carcinoid; AC: atypical carcinoid; LCNEC: large cell neuroendocrine carcinoma; SCLC: small cell lung carcinoma. ^a^ Some cases lacked follow-up information: T stage (8); N stage (8); M stage (5); Treatment QT/RT (3). ^†^ 8th International Association for the Study of Lung Cancer [1].

**Table 2 genes-13-02309-t002:** Association between EMT gene expressions and PNENs histological types by non-parametric Mann–Whitney test (*p* < 0.05).

	SCLC vs. LCNEC		*p*-value
Gene Expression (Median)	SCLC	LCNEC	
*GSK3B*	8.82	2.16	0.024
*ITGB1*	7.28	1.37	0.036
*MAP1B*	48.84	4.60	0.014
*TCF3*	14.61	1.84	0.036
*VPS13A*	15.89	3.97	0.008
*SMAD2*	7.78	2.59	0.036
	**LCNEC vs. AC**		***p*-value**
	**LCNEC**	**AC**	
*MMP3*	19.20	1.21	0.032
*ITGAV*	3.67	0.92	0.016
*KRT14*	76.11	10.55	0.016
*PLEK2*	3.97	0.25	0.016
*SNAI2*	2.17	0.38	0.032
	**AC vs. TC**		***p*-value**
	**AC**	**TC**	
*MMP3*	1.21	6.32	0.008
*SNAI2*	0.38	1.09	0.016
	**SCLC vs. AC**		** *p* ** **-value**
	**SCLC**	**AC**	
*PLEK2*	11.87	0.25	0.001
*MAP1B*	48.84	0.77	0.003
*MMP2*	2.75	0.23	0.001
*SNAI2*	4.35	0.38	0.001
*SPARC*	3.64	0.23	0.001
*WNT5A*	10.39	1.17	0.008
*ITGA5*	4.63	0.10	0.001

PNENs: Pulmonary neuroendocrine neoplasms; SCLC: Small cell lung carcinoma; LCNEC: Large cell neuroendocrine carcinoma; AC: Atypical carcinoid; TC: Typical carcinoid.

**Table 3 genes-13-02309-t003:** Interaction between EMT-related genes and Chemicals by Comparative Toxicogenomics Database (CTD).

Gene	Disease	Chemicals (Antineoplastic Drugs)—Inference Network	Inference Score
*MMP2*	Neoplasm Invasiveness	Capecitabine, Celecoxib, Doxorubicin, Tamoxifen, Sorafenib	420.88
	Neoplasm Metastasis	Capecitabine, Celecoxib, Cisplatin, Doxorubicin, Oxaliplatin, Paclitaxel, Sorafenib, Sunitinib, Tamoxifen	236.27
	Glioblastoma	Cisplatin, Doxorubicin, Paclitaxel	77.57
*MAP1B*	Cell Transformation, Neoplastic	Daunorubicin, Doxorubicin	99.91
	Lung Neoplasms	Doxorubicin	65.01
	Glioblastoma	Doxorubicin, Fenretinide	39.53
*SNAI2*	Neoplasm Invasiveness	Fulvestrant, Doxorubicin, Gefitinib, Palbociclib, Sorafenib	166.23
	Lung Neoplasms	Cisplatin, Decitabine, Doxorubicin, Erlotinib Hydrochloride, Gefitinib	155.86
	Nerve Degeneration	Cisplatin, Cytarabine, Doxorubin, Fulvestrant	127.01
*WNT5A*	Lung Neoplasms	Azacitidine, Cisplatin, Decitabine, Doxorubicin, Gemcitabine, Tamoxifen	138.80
	Cell Transformation, Neoplastic	Cisplatin, Decitabine, Doxorubicin	125.01
	Cognition Disorders	Cisplatin, Diazinon, Doxorubicin, Gemcitabine, Genistein, Methotrexate, Tamoxifen	139.07

## Data Availability

The original contributions presented in the study are publicly available. This data can be found here: https://www.ncbi.nlm.nih.gov/geo/query/acc.cgi?acc=GSE181381, accessed on 4 May 2022.

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
