# Peer review of "Proposing Specific Neuronal Epithelial-to-Mesenchymal Transition Genes as an Ancillary Tool for Differential Diagnosis among Pulmonary Neuroendocrine Neoplasms"

_genes, 2022, doi:10.3390/genes13122309_

Round 1

Reviewer 1 Report

In this manuscript, Prieto and colleagues adopted a qPCR array to profiling the expression of EMT-related genes to differentiate four entities of pulmonary neuroendocrine tumors. Their findings are interesting and may help discriminate the four subtypes of PNENs. Several issues are raised for authors to make the manuscript more comprehensive.  

1.      The procedure and challenges in the diagnosis of the four subtypes of PNENs in clinical practice could be described in more detail in the introduction section.

2.      Despite the median value of gene expression of the selected gene set statistically discriminating different subtypes of PNENs, the gene expression values of some SCLC patients could be no different from that of LCNEC patients. The situation could be seen in every comparison group in Figure 1, and this could make the subtyping of a patient unreliable based on gene expression value as when using the histology and IHC results. Are there any cutoff values for the expression value of these selected genes which could improve the discrimination power? Any new proposed strategy to improve discrimination power in clinical practice by combining the expression signature with the histology and IHC results? Please demonstrate it with your samples.

3.      The authors should make more comparisons between their findings with other earlier studies approaching the same problem using different genetic alternations such as chromosome gain/loss, CNV, and driver gene mutations or protein markers.

4.      The authors are encouraged to collect signed informed consent from patients if they are still alive.

Author Response

Reviewer 1

In this manuscript, Prieto and colleagues adopted a qPCR array to profiling the expression of EMT-related genes to differentiate four entities of pulmonary neuroendocrine tumors. Their findings are interesting and may help discriminate the four subtypes of PNENs. Several issues are raised for authors to make the manuscript more comprehensive.  

Response. First, we would like to thank the Reviewer for the critical comments that allowed us to improve the work, making it more fluent and more transparent. We revised the manuscript based on their comments and highlighted the answers to queries in yellow, and we are re-submitting the paper along with a point-by-point response to the reviewer’s comments. We are so grateful for the intellectual exercise between authors and reviewers. Undoubtedly, this exercise increased the scientific value of the work. Thank you very much.     

Query 1.      The procedure and challenges in the diagnosis of the four subtypes of PNENs in clinical practice could be described in more detail in the introduction section.

Response. We appreciate this question very much; In the new version of the manuscript, we modified the Introduction section describing with more detail about the four types of PNENs. Thank you for the suggestion.

Query 2.      Despite the median value of gene expression of the selected gene set statistically discriminating different subtypes of PNENs, the gene expression values of some SCLC patients could be no different from that of LCNEC patients. The situation could be seen in every comparison group in Figure 1, and this could make the subtyping of a patient unreliable based on gene expression value as when using the histology and IHC results. Are there any cutoff values for the expression value of these selected genes which could improve the discrimination power? Any new proposed strategy to improve discrimination power in clinical practice by combining the expression signature with the histology and IHC results? Please demonstrate it with your samples.

Response. We totally agree with the Reviewer. Thank you for drawing attention to this very pertinent observation.  In fact, re-examining the Figure 1, the gene expression values of some PNENs patients could be no different among the four types. For example,  the gene expression values of some SCLC patients could be no different from that of LCNEC patients and this gains support in the literature.  As previously reported, LCNECs were stratified into two different subgroups, “type I LCNECs” (with STK11/KEAP1 alterations) and “type II LCNECs” (with RB1 alterations) (George J, Walter V, Peifer M, Alexandrov LB, Seidel D, Leenders F, et al. Integrative genomic profiling of large-cell neuroendocrine carcinomas revealsdistinct subtypes of high-grade neuroendocrine lung tumors. Nat Commun. 2018 Mar 13;9(1):1048. doi: 10.1038/s41467-018-03099-x). Although type II LCNECs exhibit genetic similarity to SCLCs, they are distinctly from SCLCs with decreased expression of neuroendocrine markers and high performance of the NOTCH pathway (George J, Walter V, Peifer M, Alexandrov LB, Seidel D, Leenders F, et al. Integrative genomic profiling of large-cell neuroendocrine carcinomas revealsdistinct subtypes of high-grade neuroendocrine lung tumors. Nat Commun. 2018 Mar 13;9(1):1048. doi: 10.1038/s41467-018-03099-x). Interestingly, when focused on the comparison with SCLCs, the most prevalent neuroendocrine lung cancer, type I LCNECs with STK11 and KEAP1 changes displayed an increased resemblance with these carcinomas, as well as overexpression of neuroendocrine genes and a pattern of ASCL1high/DLL3high/NOTCHlow (George J, Walter V, Peifer M, Alexandrov LB, Seidel D, Leenders F, et al. Integrative genomic profiling of large-cell neuroendocrine carcinomas revealsdistinct subtypes of high-grade neuroendocrine lung tumors. Nat Commun. 2018 Mar 13;9(1):1048. doi: 10.1038/s41467-018-03099-x). Therefore, we presume that LCNEC and SCLC share EMT gene expression with clinical implications mainly related to chemotherapy response. In fact, considering only the median of the gene expression is not enough to discriminate different subtypes of PNENs. However, we try to demonstrate the gene expression of our cohort using the median expression of each gene, and the difference among the PNENs subtypes. Even though this approach may seem not reliable, we would like to thank you for pointing out this issue that helped us to realize that the use of gene expression to distinguish PNENs subtypes could be useful mainly as an ancillary tool for diagnosis, as proposed in our article, and eventually as a watershed for the treatment choice. If the Reviewer agrees, we take the liberty to include this very pertinent question in the Discussion section. Concerning the cutoff value, the data were analyzed in the StepOne software (v. 2.0, Applied Biosystems) using the Δ threshold cycle (Ct) method (2−ΔΔCt) (Livak and Schmittgen, 2001). All data were normalized by the housekeeping genes, and normal lung tissue specimens were used as case-control. The Ct cutoff was set to 35, and Fold-Change (FC) cutoff was set to ≥2.0. Thank you. 

Query 3.      The authors should make more comparisons between their findings with other earlier studies approaching the same problem using different genetic alternations such as chromosome gain/loss, CNV, and driver gene mutations or protein markers.

Response. Again, we appreciate this suggestion very much. We take the liberty to include this very pertinent question in the Discussion section. Thank you. 

Query 4.      The authors are encouraged to collect signed informed consent from patients if they are still alive.

Response. Unfortunately, in the last follow-up in 2020, only 2 patients were alive and after that, we lost contact.

Reviewer 2 Report

Table 1 clarified the number of patients who received chemotherapy. 

Can the authors clarify if patients, especially the SCLC and LCNEC ever received immunotherapy/PDL1/PD1 inhibitors? 

Table 2 summarized well the different panels. The study focused on potential panels that help differentiate between thoracic neuroendocrine tumors as MMP3 and SNA12.

Would the authors be able to suggest future directions in utilizing the knowledge of their findings for possible design of biomarker-based studies?  It seems that certain panels have the potential to benefit from PARP inhibition through the DNA damage mechanism.

Author Response

Reviewer 2

Response. First, we would like to thank the Reviewer for the critical comments that allowed us to improve the work, making it more fluent and more transparent. We revised the manuscript based on their comments and highlighted the answers to queries in yellow, and we are re-submitting the paper along with a point-by-point response to the reviewer’s comments. We are so grateful for the intellectual exercise between authors and reviewers. Undoubtedly, this exercise increased the scientific value of the work. Thank you very much.     

Query 1. Table 1 clarified the number of patients who received chemotherapy.

Response. Thank you for this comment, in the manuscript we describe the number of patients who received chemotherapy as follows:

Systemic Chemotherapy                                          

Yes      8 (80.0%)        3 (75.0%)        2 (50.0%)        2 (66.7%) = 15 patients received chemotherapy

No       2 (20.0%)        1 (25.0%)        2 (50.0%)        1 (33.3%) = 6 did not receive chemotherapy

Note. 3 cases lacked follow-up information.

Query 2. Can the authors clarify if patients, especially the SCLC and LCNEC ever received immunotherapy/PDL1/PD1 inhibitors?

Response. Thank you for this question, however in our cohort no patient underwent the immunotherapy treatment.

Query 3. Table 2 summarized well the different panels. The study focused on potential panels that help differentiate between thoracic neuroendocrine tumors as MMP3 and SNA12.

Response. Thank you for this comment. Yes, in this study we try to analyze the potential markers that could be helpful in distinguishing the PNENs subtypes.

Query 4. Would the authors be able to suggest future directions in utilizing the knowledge of their findings for possible design of biomarker-based studies?  It seems that certain panels have the potential to benefit from PARP inhibition through the DNA damage mechanism.

Response. We appreciate this comment very much. Therefore, we take the liberty to add this information to finalize the discussion. PNENs are known for their histology overlapping and clinical behavior. The interesting point about those is the suggestion that differences in behavior and response to therapy occur because of the low and high epithelial-to-mesenchymal (EMT) gene expression in the four major types of PNENs, and these differences can be a key to improving the diagnosis and response to treatment. In recent years, there has been a great evolution of Pathology through the implementation of diagnostic techniques that minimize the inherent subjectivity of the Pathologist, improving the accuracy of diagnosis of malignant and nonmalignant diseases. Here, we propose the use of quantitation of EMT genes expression to characterize tumor cell populations, and eventual response to treatment.  In the emergent scenario, for example, PARP inhibition through DNA damage mechanism can be implemented as a diagnostic and predictive biomarker for the selection of patients for personalized treatments.

Round 2

Reviewer 1 Report

The authors have responded to the concerns raised in the previous version of their manuscript. Despite some limitations remaining, the collected materials and information could be a good basis for them and other researchers to further explore useful markers to help classify different PNEN subtypes.